# Evidence-based practice and its associated factors among point-of-care nurses working at the teaching and specialized hospitals of Northwest Ethiopia: A concurrent study

**Abebe Birhanu Degu**[1]*, **Tesfahun Melese Yilma**[1], **Miftah Abdella Beshir**[1], **Anushia Inthiran**[2]

**1** Department of Health Informatics, College of Medicine and Health Sciences, University of Gondar, Gondar, Ethiopia, **2** Department of Accounting and Information Systems, College of Business and Law, University of Canterbury, Meremere, New Zealand

* abebirhanu21@gmail.com

## Abstract

Evidence-based practice (EBP) is the application of the best scientific evidence for clinical decision-making in professional patient care. In Ethiopia, the majority of nursing care is based on experience, tradition, intuition, common sense, and untested hypotheses. Evidence-based clinical practice has the potential to increase the quality of healthcare services while also lowering costs and increasing clinical outcomes. An institutional-based concurrent study design method of quantitative and qualitative research was conducted from Feb. 30 to Apr. 20, 2020. Systematic random sampling and purposive sampling techniques were used to select the study participants for the quantitative and qualitative analyses, respectively. To collect quantitative data, a pretested, structured, and self-administered questionnaire was used, and to collect qualitative data, an in-depth interview guided by questions was used. EBP was found to be significantly associated with educational level (AOR = 2.15, CI (1.15–4.02)), administrative support for EBP (AOR = 1.89, CI (1.22–2.91)), attitude toward EBP (AOR = 1.80, CI (1.24–2.62)), and preference of available information sources (AOR: 2.32, CI (1.58–3.39). The four main themes that emerged from the conventional content data analysis were the advantages of EBP application, barriers to EBP implementation, enabling factors for EBP, and evidence sharing. According to the findings of this study, only a few nurses used EBP at a high level. The implementation of EBP was positively associated with educational level, attitude toward EBP, administrative support, and the availability of information resources, as confirmed by the qualitative study. There must be an intervention program to facilitate the implementation of evidence in nursing practice by stakeholders to improve and increase the efficacy of practicing EBP.

**Data Availability Statement:** All data will be available at the DOI:10.5281/zenodo.6299842 without legal or ethical restriction in data sharing.

**Funding:** The author(s) received no specific funding for this work

**Competing interests:** The authors have declared that no competing interests exist.

**Abbreviations:** AOR, stands for adjusted odds ratio, CI stands for confidence interval, and COR stands for the crude odds ratio; EBP, Evidence-Based Practice; G ADM, A group of nurses with administrative responsibilities; G TEAM, A group of nurses with team leader responsibilities; G STA<5, Staff nurses with less than 5 years of professional experience; G STA5–10, Staff nurses with 5 to 10 years of professional experience; G STA > 10, Staff nurses with more than 10 years of professional experience.

# Introduction

Evidence-based practice (EBP) is the application of the best scientific evidence in clinical decision-making by integrating clinical experience, incorporating patient values and preferences into the practice of professional patient care [1]. It is about making decisions by using the best available evidence from multiple sources in a conscientious, explicit, and judicious manner [2].

Evidence is data derived from historical or scientific evaluations of procedures that can be used by healthcare decision-makers. Experimental (randomized clinical trials, meta-analyses, and analytic studies) and non-experimental (quasi-experimental, observational) evidence, as well as expert opinion and historical or experience evidence, are all valid types of evidence [3].

Nursing is a science that draws on study findings to make decisions. Scientific research is the gold standard for deriving information from science. Nursing practice generates research questions, whereas research is the foundation of current practice. Therefore, practice and research exist in a circular continuity beside one another [4].

EBP, which is the use of theoretical research-based findings in conjunction with reliable forms of evidence in clinical decision-making, is currently essential in nursing practice to promote optimal patient outcomes by incorporating research findings, clinical experience, and patient preferences [5].

EBP has been hailed as the gold standard for providing compassionate and safe care while also encouraging nursing excellence. Despite the ever-increasing availability of healthcare information and constant government pressure, the use of research-based evidence in nursing practice is restricted; patient care is impacted by the experiences and opinions of those involved in treatment. Across the board, studies reveal that interventions have failed to be both successful and cost-efficient [6].

In low- and middle-income nations, EBP is not frequently used. For many healthcare institutions, EBP procedures are a relatively new and frequently intimidating task. In Africa, for example, EBP is stressed and encouraged for nurses in countries such as South Africa, Ethiopia, Kenya, Nigeria, Egypt, Botswana, Burundi, and Malawi [7–10]. However, EBP in nursing practice is still in its early stages. For example, according to a recent study from Nigeria, EBP is not commonly used in the country's healthcare system [7]. In Africa, EBP is still a challenge. One explanation for this difficulty is that Africa falls behind in research due to a paucity of studies describing the state-of-the-art in EBP and a lack of government financing [10]. Another cause could be a lack of resources, which makes it nearly impossible for health professionals to work with vulnerable groups in low-income settings to acquire information [11].

Nurses may lack the skills or experience needed to extract research evidence from the literature or to apply that evidence, making it ineffective and failing to enhance clinical outcomes. Transferring research into clinical practice is fraught with difficulties, and the adoption process is difficult [12]. Such a situation produces medical errors such as misdiagnosis, incorrect treatment, an increase in multidrug resistance, severe iatrogenic injuries, and unexpected patient fatalities are all prevalent medical errors in today's health care systems [13, 14].

The nursing staff is the largest health professional group in all health care sectors, working indirect patient care, assessing patients' needs, and making long-term choices on nursing interventions [15]. However, at all levels of the health system, there is little culture or tradition of conducting, trusting, or using EBP by nurses [16].

Even though several studies have attempted to analyze EBP from various perspectives, the detailed factors are not analyzed, particularly in developing countries such as Ethiopia and also in the study area, as professional development and information and communication technologies (ICT) becomes more advanced. If nurses use EBP to improve their continuous education,

it will benefit their ability to improve the quality of care they provide to patients. Furthermore, it is unknown how much health-related research is being used to improve healthcare practice to ensure high-quality healthcare delivery and improve quality of life standards. As a result, the purpose of this study was to evaluate evidence-based practice and its associated factors among nurses working in the study settings.

## Materials and methods

### The research design and setting

An institution-based concurrent study design using quantitative and qualitative methods was conducted from February 30 to April 20, 2020. This research was carried out in the two governmental teaching and specialized hospitals of the Amhara Regional State, Northwest Ethiopia (the University of Gondar and Tibebe Gion), serving 3.5–5 million people each. The hospitals are the only teaching and specialized hospitals in the Amhara region. They were selected because:

They serve as professional development centers for both in-service and out-of-service health professionals, as well as undergraduate and postgraduate medical students, have a reputation for research excellence as a larger referral center in the region because they are staffed with the most senior experts who handle patients with complex illnesses and are exposed to highly specialized treatment with well-resourced diagnostic and therapeutic facilities, have strategic objectives to generate evidence-based healthcare professionals who are committed to ongoing professional development and better patient care and nurses are expected to use EBP because, in addition to providing daily patient care, they also serve as practical educational venues for students and healthcare workers.

### Research participants

All nurses with a Bachelor of Science (BSc) or higher in nursing who worked in the study settings were included in this study.

This study excluded nurses on leave (annual, maternal, and/or sick leave), with less than six months of work experience, nurses in postgraduate programs, and nurses on academic staff because postgraduate program followers and students were not eligible during the data collection period, and nurses on academic staff were thought to have better EBP experience in their academic activities [17], which skewed the results of the study.

### Sample size determination

Both the single and double population proportion formulas were used to calculate the sample size.

The formula for calculating a single population proportion

$$n = \frac{(z_{\alpha/2})^{2*}p(1-p)}{(d)^2}$$

Where z = 95% confidence level, d = 5% margin of error, and 51.8% proportion (p) of EBP from the previous study [16]. Assuming a 10% non-response rate, the total sample size became 423.

The formula for doubling the population proportion is

From the previous study [16], we calculated the sample size for each factor variable. A power of 80%, a confidence level of 95%, and an odds ratio of 1:1 (see Table 1 below).

**Table 1. A calculated sample size of EBP by Epi Info of each factor variable from the previous cross-sectional study.**

| variables | category | EBP use | EBP not use | Prop | COR | sample size |
|---|---|---|---|---|---|---|
| Sex | Male | 86 | 53 | 61.9 | 2.488 | 172 |
| | Female | 45 | 69 | 39.5 | | |
| Educational Level | BSc | 107 | 61 | 63.7 | 4.458 | 72 |
| | Diploma | 24 | 61 | 28.2 | | |
| Current Role | Head Nurse | 15 | 4 | 78.9 | 3.815 | 96 |
| | Staff Nurse | 116 | 118 | 49.6 | | |
| Knowledge | Knowledgeable | 95 | 61 | 60.9 | 2.639 | 154 |
| | Not Knowledgeable | 36 | 61 | 37.1 | | |
| Lack of Autonomy to change practice | Neutral | 21 | 24 | 46.7 | 1.718 | 482 |
| | Agree | 27 | 53 | 33.8 | | |
| Inability to properly interpret the result of research | Neutral | 27 | 19 | 58.7 | 4.895 | 66 |
| | Agree | 18 | 62 | 22.5 | | |

Taking the largest sample size, 482, and assuming a non-response rate of 10% (482) = 48, the final sample size was calculated to be 482+48 = 530.

## Sampling procedures

Participants were chosen proportionally from the two hospitals for the study. The hospital nurse lists were utilized as a sampling frame for selecting potential study participants in the sample. For all nurses working at the same hospital, EBP was believed to be uniform. The study's participants were then chosen at random from the sample frame using record identification numbers. The lottery method was used to randomly pick nurses as study participants (Fig 1).

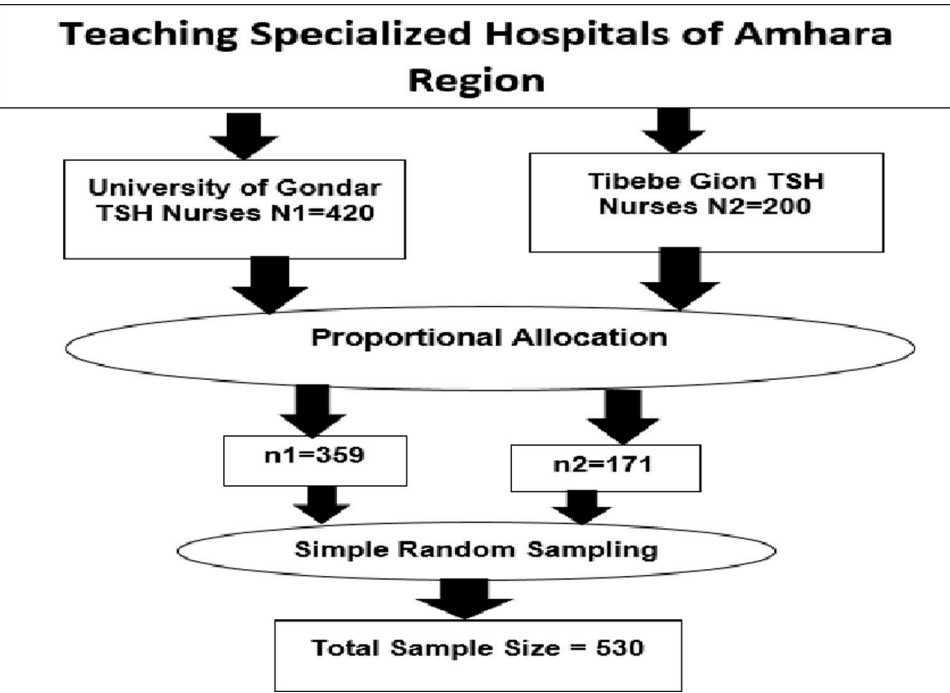

**Fig 1. Sampling procedure of the study of evidence-based practice among nurses at the University of Gondar and Tibebe Gion teaching specialized hospitals, 2020.**

Among 530 candidate study participants, a total of 507 BSc and above nurses who returned the questionnaire were taken from 10 administrative nurses, 34 nurses' team leaders, and 463 staff nurses by simple random sampling.

Twelve key informants were chosen at random for an in-depth interview till information saturation was achieved. Professional and practical experience (clinical staff and/or management) were both considered when choosing key informants. Working in a hospital for at least the past six months was the initial selection criterion. Nurses were classified into four groups based on whether they met these criteria: These nurses were further subdivided into the following groups: 1. Group of Administrative Nurses (G ADM) 2. Team leader nurses group (G TEAM) and 3. Clinical Staff nurses: additionally, these nurses were subdivided into three groups:

 3.1. No more than five years of hospital experience (G STA5)

 3.2. 5–10 years of hospital experience (G STA5–10)

 3.3. More than 10 years of experience in the hospital (G STA > 10).

## Data collection tools and procedures

Data were collected using a pretested and structured self-administered questionnaire, the majority of which were adapted from previous Ethiopian studies [10, 16, 18], and some of which were adapted from other studies [19, 20] (see S1 Appendix). The questionnaire was written in English because the study participants were nurse professionals with a bachelor's degree or higher who were thought to be fluent in English. The guiding questions for the in-depth interview were written in English and translated into the local language (Amharic) for simplicity, validity, and clarity.

Furthermore, the questionnaire was pretested at Felege Hiwot Referral Hospital with 5% of the sample size, and questionnaire reduction, jargon word elimination, grammatical correction, and other changes were made. Cronbach's alpha reliability test (0.91) was used to assess the tool's reliability for each subscale, indicating that the questionnaire's parameters were sufficiently reliable.

One BSc nurse and one health informatics professional collected quantitative data from nurses by distributing structured self-administered questioners to the nurses after explaining the purpose and technique for filling out the questionnaire. The Principal Investigator (PI) has done continuous monitoring and supervision throughout the data collection period.

The PI gathered qualitative data from key informants. Purposive convenience sampling was used to choose key informants for in-depth interviews. The duration of each in-depth interview was 30–35 minutes. Information from key informants was recorded using notes and audio recorders.

The questionnaire of the quantitative study consisted of eight sections: about EBP, Knowledge, Preferences/Use of available information sources, Awareness about Electronic Information Sources, Information Searching Skills, Nurses' Self-Efficacy, Attitude, and Factors associated with Evidence-Based Practice. Each section had several subscales with greater than seven questions made by different researchers to measure it.

To assess EBP, seven self-reported questions were asked. Each question had a four-point response scale (never, monthly, weekly, and daily). These categories were assigned numerical values ranging from 1 to 4 (Never '1', Monthly '2', Weekly '3', and Daily '4'). As a result, the respondent received an EBP total score (from a minimum of 7 to a possible maximum of 28). Each question's average can be calculated as $(1+2+3+4)/4 = 2.5$. A score of 2.5 or higher was considered a desirable response to the EBP questions. The average score of the seven questions was used to assess EBP. As a result, $2.5 \times 7 = 17.5$ is chosen as the cut-off point. Respondents

with a total EBP score greater than 17.5 were considered to have "Good EBP" while those with a score less than 17.5 were labeled as having "Poor EBP."

In total, ten questions were asked to assess the extent of EBP knowledge, with Yes, No, and I don't know (IDK) answers. Yes, the responses are coded as "1", "no", and "0" for I don't know (IDK). For each respondent, a cumulative EBP knowledge score (ranging from 0 to 10) was calculated. The mean knowledge score was calculated after checking for normal distribution to test the knowledge score using the Shapiro–Wilk test (W = 0.982 and p = 0.11), and a visual inspection of their histogram, normal Q-Q plots, and box plots revealed that knowledge scores were approximately normally distributed with a skewness of-0.167 (SE = 0.218), a kurtosis of-0.247 (SE = 0.433), and a mean of 6 (SD = 1.73). Respondents having a knowledge score above the mean value are considered to have "good knowledge" unless they have "poor knowledge."

Ten questions were used to determine the level of attitude toward EBP. Every attitude question had a five-point Likert scale (strongly agreed coded as 5, agreed coded as 4, neutral coded as 3, disagreed coded as 2, and strongly disagreed coded as 1). Each respondent had a minimum attitude score of 10 points and a maximum attitude score of 50 points. The Shapiro Wilk test was used to check for normal distribution of an EBP score (W = 0.983, p = 0.589), and a visual inspection of their histogram, normal Q-Q plots, and box plots revealed that attitude scores were approximately normally distributed with skewness of 0.015 (SE = 0.316), kurtosis of -0.206 (SE = 0.623), and mean of 36 (SD = 3.921). Respondents with attitudes toward evidence-based practice that was higher than the mean score were classified as having a "favorable attitude" while those with attitudes that were lower were classified as having an "unfavorable attitude".

Six questions on a five-point Likert scale were used to assess respondents' skill in information searching (finding, reviewing, and using various sources of evidence) (very good coded as 5, well coded as 4, neither good nor poor coded as 3, poorly coded as 2, and very poor coded as 1). A total EBP skill score (from a minimum of 6 to a maximum of 30) was calculated for each respondent. The mean skill score was calculated after checking for normal distribution to test the skill score using the Shapiro–Wilk test [21] (W = 0.971, p = 0.09) and a visual inspection of their histograms, normal Q-Q plots, and box plots showed that skill scores were approximately normally distributed with skewness of -0.124 (SE = -0.279), a kurtosis of -0.206 (SE = 0.552), and a mean of 19 (SD = 2.968) and labeling respondents having skill scores above the mean value as "good skill" unless they had a "poor skill".

On a five-point Likert scale, seven questions were provided to assess the use of available information sources (never "1", rarely "2", sometimes "3", often "4", and always "5"). The Shapiro–Wilk test [21] (W = 0.945, p = 0.79) was used to test the use score, and a visual inspection of their histogram, normal Q-Q plots, and box plots revealed that preference scores were approximately normally distributed with skewness of -0.249 (SE = 0.398), kurtosis of 0.335 (SE = 0.495), and a mean of 12.8 (SD = 0.474) and the labeling of respondents having use scores above the mean value as "preferred available information source" unless it is marked as "not preferred".

On a three-point Likert scale, eleven questions were provided to assess awareness of electronic information sources (unaware "1", aware but not used in clinical decision making "2", aware and used "3"). The mean awareness score was calculated after testing for normal distribution with the Shapiro–Wilk test [21] (W = 0.969, p = 0.197), and a visual inspection of their histogram, normal Q-Q plots, and box plots revealed that awareness scores were approximately normally distributed with skewness of -0.004 (SE = 0.333), kurtosis of -0.368 (SE = 0.656), and a mean of 14 (SD = 0.487) and labeling respondents having awareness scores above the mean value as "aware" electronic information source unless they were "not aware".

On a three-point Likert scale, seven questions were provided to assess nurses' self-efficacy (not confident "1", moderately confident "2", and confident "3"). The mean self-efficacy score was calculated after testing for normal distribution using the Shapiro–Wilk test [21] (W = 0.981, p = 0.258), and a visual inspection of their histogram, normal Q-Q plots, and box plots revealed that preference scores were approximately normally distributed with skewness of 0.172 (SE = 0.267), kurtosis of 0.302 (SE = 0.529), and a mean = 14.72 (SD = 0.307) and the labeling of respondents having self-efficacy scores above the mean value as "confident" unless they were "not confident".

## Management and analysis of data

The data were initially coded and entered into Epi-data version 4.6.0 (Epi-Data Entry is used for simple or programmed data entry and data documentation). It handles simple forms or related systems. Optimized documentation and error detection features) and exports them to SPSS version 20 for analysis. Data analysis includes descriptive statistics to describe the demographic characteristics of the participants, and the data is described using text, tables, and graphs. A chi-square test was done for all categorical independent variables. A multi-co linearity test was used to check the variance inflation factor (VIF). Binary logistic regression was used to determine the association between the outcome variables and predictors. Statistical tests at 95% CI and a P-value of 0.05 were conducted. Variables having a p-value of up to 0.2 in the bi-variable analysis were selected to fit the model in the multivariable analysis. Finally, a P-value of 0.05 in the multivariable model of logistic regression was used to select variables that had a statistically significant association with EBP.

The qualitative data were transcribed and interpreted in English after being recorded in Amharic. An inductive coding approach was used to conduct a thematic analysis. The coding technique was carried out by the researcher to assure the analyses' rigor and credibility. The researcher recognized emergent subjects and trends in an initial review of the data using this triangulation technique and drew up a list of prospective topics that were then utilized to set up the codes using the Lundman and Graneheim data analysis approach. The data was reduced by matching codes and establishing relationships and analogies. The Open Code software aided in the qualitative data analysis.

A technique of sustained interaction with participants, persistent observation, peer-debriefing, and member-checking was used to establish trustworthiness. Iterative questioning of the facts and return to study it numerous times were also used to present evidence.

## Ethical consideration

Ethical clearance was obtained from the Institutional Review Board of the University of Gondar. Official letters of support were presented to the study settings, the intentions and grandness of the subject were explained, and each participant provided verbal informed consent. Participants were informed that all data obtained from them was maintained confidentially and anonymously by using codes instead of personal identifiers.

## Results

### Socio-demographic characteristics

A total of 530 questionnaires were distributed to the study participants, with 507 of the questionnaires being returned (a response rate of 95.7%). Of the returned questionnaires, 340 (67.1%) and 167 (32.9%) were from the University of Gondar and Tibebe Gion teaching specialized hospitals, respectively. The respondents were 273 (53.8%) female and 234 (46.2%)

**Table 2. Socio-demographic characteristics of nurses in the University of Gondar and Tibebe Gion teaching specialized hospitals, Ethiopia 2020 (n = 507).**

| Variables | Categories | Male (%) | Female (%) |
|---|---|---|---|
| Age_catagories | 20–24 | 2.6 | 3.6 |
| | 25–29 | 20.5 | 27.4 |
| | 30–34 | 15 | 18.7 |
| | ≥35 | 4.9 | 7.3 |
| Educational Level | BSc Degree | 37.9 | 50.3 |
| | MSc Degree | 5.1 | 6.7 |
| Work Experience | 5< | 21.7 | 27 |
| | 5–10 | 18.3 | 23.9 |
| | >10 | 3 | 6.1 |
| Current place of work | UoG teaching Specialized Referral hospital | 28 | 39.1 |
| | Tibebe Gion teaching Specialized Referral hospital | 15 | 17.9 |
| Working Unit | Inpatient wards | 25.2 | 28.8 |
| | ICU wards | 3.7 | 6.5 |
| | OR | 3.2 | 6.1 |
| | Emergency wards | 6.5 | 5.5 |
| | Outpatient department | 4.3 | 10.1 |
| Work position | Staff nurse | 38.3 | 53.1 |
| | Team leader Nurse | 3.9 | 2.8 |
| | Admin nurse | 0.8 | 1.2 |
| Training related to EBP | No | 31.4 | 43.4 |
| | Yes | 11.6 | 13.6 |
| Electronic device used | Low-end phones or basic phones | 8.3 | 7.3 |
| | Feature phones or internet-enabled phones | 10.3 | 16.4 |
| | Smartphone | 14 | 24.1 |
| | Tablet computer | 1 | 0.4 |
| | Laptop Computer | 5.1 | 6.1 |
| Internet access at home | No | 45.7 | 45.7 |
| | Yes | 4.3 | 4.3 |

male. Of those, 463 (91.3%) were staff nurses, 34 (6.7%) were team lead nurses and 10 (2.0%) were administrative nurses. Of the 507 study subjects, the mean age was 29.71 (SD 4.42). Most of the subjects in the study were BSc, which accounts for 447 (88.2%). Only 128 (25.2%) got training related to EBP from 68 (13.4%) undergraduate and 60 (11.8%) postgraduate. The detailed information is provided in Table 2.

## Evidence-based practice in the study area

Of the total 507 nurses, 239 (47%) [95% CI of 43% to 51%] had incorporated EBP into their clinical practice. Their practice rate was assessed with three options (sometimes/monthly, usually/weekly, and always/daily) by self-reporting. Only 160 (32%), 216 (43%), and 76 (15%) use EBP in their clinical practice on a monthly, weekly, and regular basis, respectively (Fig 2).

**Knowledge about evidence-based practice.** Once asked how familiar the respondents were with the concept, about 441 (87%) were familiar with the concept of EBP, with a mean knowledge of 6 (SD 12.9). When the respondents were asked questions about research terms, about 217 (42.8%) of respondents scored low, and about 290 (57.2%) of respondents scored high, with 58% [95% CI of 56.2% to 59.2%] of overall knowledge.

**Preferences/use of available information sources for evidence-based practice.** We found that human information sources (such as sharing with colleagues), using medical

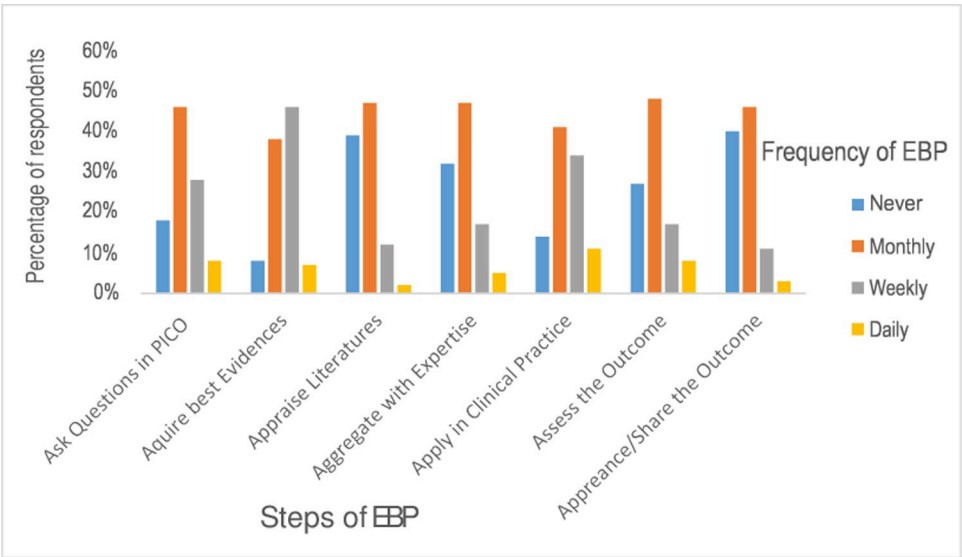

**Fig 2. Evidence-based practice among nurses at the University of Gondar and Tibebe Gion teaching specialized hospitals by nurses in 2020.**

applications, and policies and guidelines were used more frequently by them. It is worthy of notice that the use of electronic information sources (medical journals and databases) among nurses was the least common (Fig 3).

**Awareness about electronic information sources for evidence-based practice.** The majority of nurses are relatively aware and use medical applications such as Medscape rather than the different medical journals and databases of electronic information sources as seen in Fig 4.

**Information searching skills for evidence-based practice.** Most nurses (60%) are unlikely to skim down or quickly browse resources. There are some problems, such as a lack of

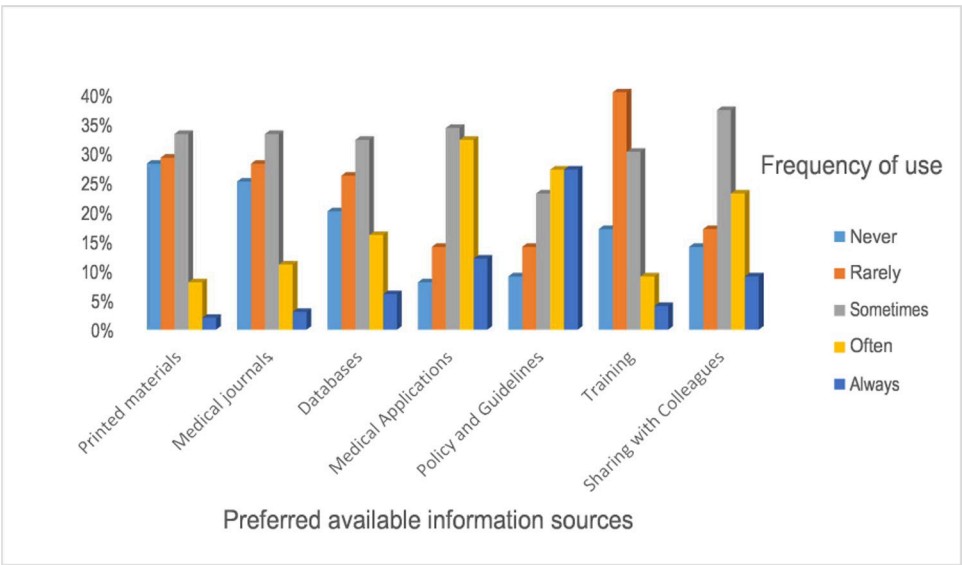

**Fig 3. Preferences on sources of evidence-based practice among nurses in University of Gondar and Tibebe Gion teaching specialized hospitals, 2020 (n = 507).**

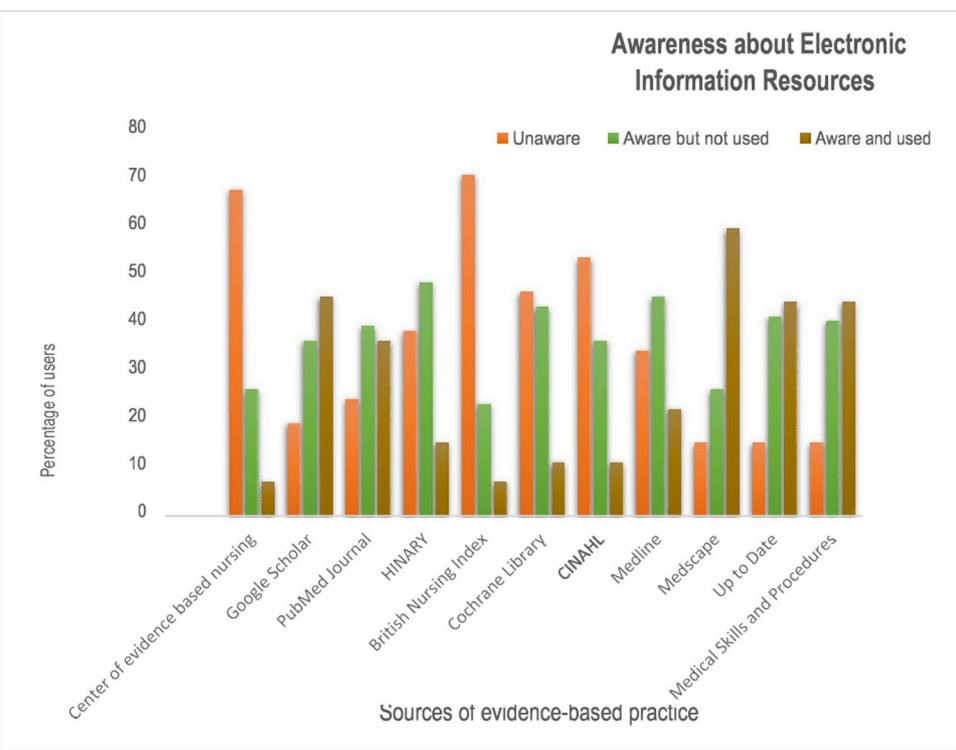

**Fig 4. Awareness about electronic information sources of evidence-based practice among nurses in University of Gondar and Tibebe Gion teaching specialized hospitals, 2020 (n = 507).**

information technology skills, access to research articles, and a lack of references in the native language because most research is published in a foreign language.

Even though participants had overall information searching skill of 64%, it is worth remembering that several nurses have revealed that they are not comfortable with Boolean and proximity operators (54%) (Fig 5).

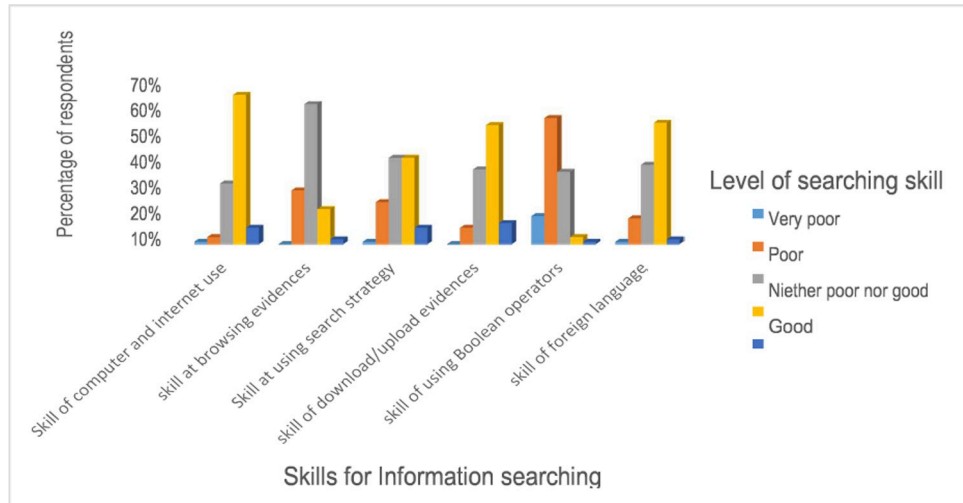

**Fig 5. Information searching skill for evidence-based practice in University of Gondar and Tibebe Gion teaching specialized hospitals, 2020 (n = 507).**

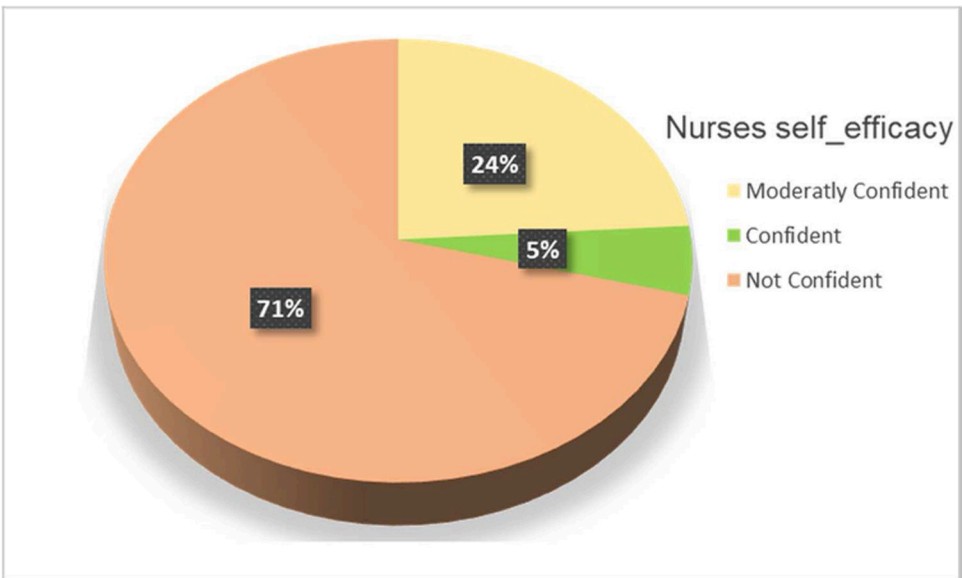

**Fig 6. Nurses self-efficacy in evidence-based practice at the University of Gondar and Tibebe Gion teaching specialized hospitals, 2020 (n = 507).**

**Nurses' self-efficacy in evidence-based practice.**   The majority (71%) of nurses have no confidence, but only 5% of nurses have confidence in their decision-making in day-to-day clinical practice (Fig 6).

**Attitude towards evidence-based practice.**   Respondents had a mean attitude score of 36 (SD 3.92). However, fifty-five percent of respondents had a favorable attitude towards EBP. Based on the findings, 71% of respondents agreed that research and literature findings are useful in day-to-day practice, 58.5% of respondents agreed that EBP improves the quality of healthcare, and 23.1% agreed that their reimbursement rate would increase if they incorporated EBP into their practice (Fig 7).

**Factors associated with evidence-based practice.**   When asked about barriers that impeded their EBP, 52% of the participants reported that there were barriers to EBP implementation. The identified barriers include: availability of information sources 16% (n = 41), computer access 89% (n = 235), internet access 89.7% (n = 237), internet speed 53% (n = 140), administrative support 74% (n = 195), search strategy and EBP training 80% (n = 211), enough time for practice 56% (n = 148), awareness about EBP 58% (n = 153), understanding literature in a foreign language 28% (n = 74), physicians' cooperation 55% (n = 145), understanding statistical terms 54% (n = 143), and literature with conflicting reports 66% (n = 174) and lack of resources 76% (n = 201). Some respondents identified over one barrier.

## Analyzing qualitative outcomes

For an in-depth interview, 12 BSc and above (7 staff, 3 team leaders, and 2 administrative) nurse key informants were selected purposively from 113 BSc and above nurses who were not selected for quantitative until the data was saturated.

Four main themes arose from the conventional content data analysis: (1) benefits of implementing EBP (2) impediments to EBP implementation (3) EBP enablers (4) The exchange of evidence.

**1. Advantages of the application of EBP.**   This theme addressed what participants saw as a positive outcome when relying on data for research and clinical practice, as well as how EBP reduces uncertainty.

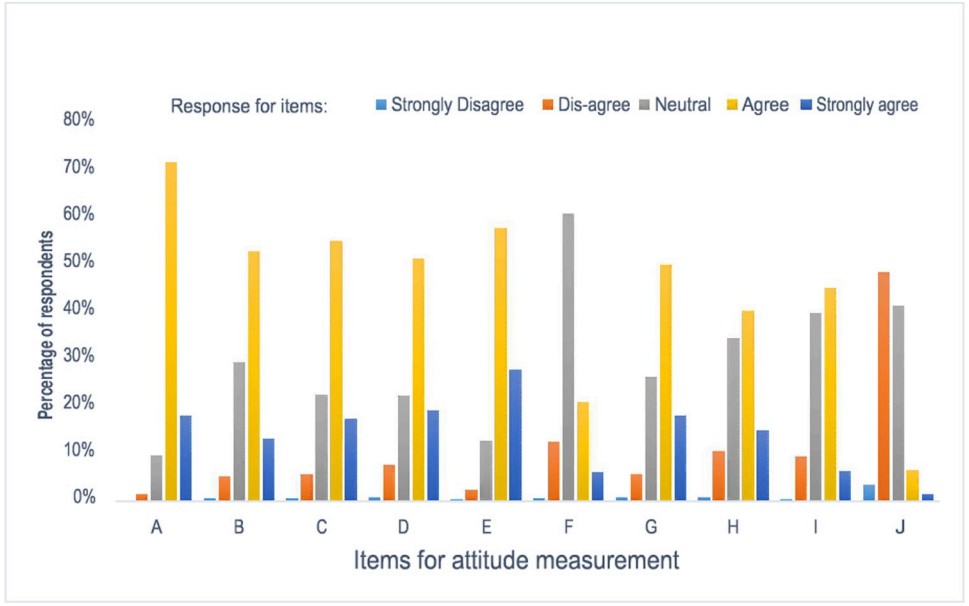

**Fig 7. Self-reported attitudes towards evidence-based practice among nurses in the University of Gondar and Tibebe Gion teachings specialized hospitals, Ethiopia 2020.**

*"I believe we all operate in the same manner to provide the best healthcare."* (G ADN) (M1)

They also believed that evidence-based healthcare provided better care for patients and made better use of available resources and time:

*"I think you'll give more appropriate treatment to patients if you base yourself on the facts. With fewer resources, you can produce better outcomes."* (M1) (G TEAM)

Participants claimed that scientifically supported clinical practice enhances the professional image and provides legal support.

*"I believe that it makes the profession more valuable and reinforces it. Then, it is a back-up when you say. Well, we're working on a scientific method."* (G TEAM) (M2)

They described the EBP as encouraging, as well as a challenge and motivation.

*"Yes, I believe so as well. When you see other people doing things, it makes you want to do the same. When you go to another place where people are settled, they don't care about anything and just go to work."* (F1) (G STA greater than 10)

*"I would love to base the evidence on all my work. I would be happy if I could investigate at work. . ."* (F2) (G STA 5–10)

EBP leads to an awareness of what it means to be a nurse and what that nurse's professional responsibility is. Also, EBP entrusts the patient with a professional commitment.

*"Yet, where do our shortcomings lie? We have to know them and know when we are doing our job properly."* (M1) (G STA 5–10)

*"Then, I have to do research, I have to answer this person who is coming and telling me something about tuberculosis. . ."* (M1) (G TEAM)

**2. Barriers to the application of EBP.** Obstacles to implementing EBP have arisen more frequently than benefits from its implementation. One participant expressed concern about the lack of incentives to encourage and recognize EBP, emphasizing the importance of institutional support and motivation:

*"In a way, I believe it should be the responsibility of nursing administration to motivate, encourage, and generate interest in nurses, but I'm not sure how." "I believe that training should come first, followed by encouragement."* 5–10 (G STA)(F2)

*"We clinical staff require the support of our higher administrators in conducting clinical research because we know our problem better than academic staff, and if we have the opportunity to investigate our clinical problem, we can integrate research evidence into our daily practices."* (G STA 5–10) (M2)

Nurses do not agree that variables such as finances, patient type, or professional characteristics are taken into account when designing protocols or clinical practice guidelines.

*"To start the protocol process, do we take into account the needs of each population group with which we work? This doesn't seem so, as it takes them into account when applying."* (STA G greater than 10)(M3)

*"A training center has to be established to give us training on how to integrate research evidence into our practices and to develop guidelines for practice, thus serving as a source for guiding our health interventions."* (F3) (G STA5)

Participants discovered that the clinical procedure is opposed by some health practitioners. They discussed the burden placed on them, the patients, and their colleagues by hospital specialists, as well as the lack of written scientific documentation:

*"The patients who push and tell you another nurse healed them the previous day, they did so differently, and they are the ones who force you to do so."* (F1) (G STA > 10)

*"Throughout, obtaining information and knowledge-based preparation are not provided, and there is a lack of understanding and provision that the practitioner can work through the facilitator throughout developing a documentary environment."* (F4) (G STA 5–10)

Participants were also exhausted and dissatisfied with institutional assistance in their efforts to bring about more sound hospital improvements:

*"We have just attended a conference, haven't we? Obscure things have we known, have we not? Then you send a session and say, look, it is explained here, not proof-based. The next day, you find out that they continue to do the same. It doesn't matter. . ."*(G STA 5–10) (M1)

Doubts about new evidence and resistance to change were the most frequently mentioned obstacles. All interviewees offered weak excuses for not using EBPs, such as their complexity and a lack of appropriate clinical resources and studies.

*"Now that we are attempting to implement EBP, yes, it is very complicated to constantly wonder if you are doing it correctly."* 5–10 (G STA) (M2)

*"No hospital library, revised rules, inspiration, adequate training, and computers to enable nurses to upgrade."* (F2) (G STA 5–10)

Participants also expressed reservations about using the findings in clinical practice.

*"Instead, I believe the scientific method is excellent, providing a career and all, but I do not believe we must become robots simply because it has been proven."* (M3) (G STA > 10)

Participants agreed that, while a theoretical foundation supports the new approach, changing established norms is extremely difficult. This segment was thoroughly addressed in every interview, particularly in the interview with less experienced staff:
It is difficult to change the first problem you face. People say to you: *"No, I always did so, so it works well."* 5–10 (G STA) (M2)
They also mentioned an inadequate knowledge of EBP methodology.

*"The first challenge is adapting, then, and preparing so that I can do it. And then, it's doubt, what if I do something wrong? What if I can't do that?"* (M1) (G ADM)

**3. Enabling factors for EBP.** This theme included what participants regarded as an EBP facilitator.

*"Even if it is insufficient, some nurses are occasionally trained, and the Internet is made available to all professionals, as well as training on information sources and methods of use. Nurses are unable to provide services based on a patient's informed choice due to insufficient training and inadequacy of authority to change procedures of care."* 5–10 (G STA) (F2)

*"A training center has to be established to give us training on how to integrate research evidence into our practices and to develop guidelines for practice, thus serving as a source for guiding our health interventions."* (F3) (G STA5)

**4. Sharing of evidence.** Concerning how new evidence is shared, all participants agreed that they do not receive formal working time during which they are taught by non-hospital experts as well as hospital nurses who prefer to participate in the work:

*"In the sessions, we try to share it. For example, one of our colleagues has held an injection workshop because he knows that"* (F4) (G STA5-10)

*"For if you say that I know nothing about it and that someone comes and does it, it means that you don't care what you do."* (M3) (G STA > 10)

In situations and contexts not specifically designed to share scientific data, EBP has occasionally been promoted.
*It's usually in the tea break many times when we say, "Remember what we talked about a few days ago? Do you remember? That's a scientific paper I've found. . ."* (M1) (G TEAM)
Participants further discussed variations in clinical quality and quantity and other methods of EBP sharing between hospitals. *"The difference is so vast that I have been working in several health centers and hospitals."* (F1) (G STA > 10)

## Discussion

The purpose of this study was to assess evidence-based practice and its associated factors among nurses working at the University of Gondar and Tibebe Gion Teaching Specialized Hospitals in the Amhara Region, Northwest Ethiopia.

The findings of this study revealed a low prevalence of EBP, and the associated factors eligible for EBP were educational level, nurses' attitudes toward EBP use, administrative support, and preferred available information sources.

In this study, EBP among nurses in the study settings was found to be 47%. Only 207 (41%), 174 (34%), and 57 (11%) apply/use EBP in their clinical practice monthly, weekly, and daily, respectively. This finding was consistent with the study done in Tikur Anbesa hospital in Addis Ababa, in which 57.6% of participants applied EBP. Of them, 64 (52.8%), 38 (31.4%), and 19 (15.7%) applied EBP sometimes, usually, and always, respectively [10]. This contradicted the findings of Nigeria [9] where 55.5% and 30.9% of nurses used EBP occasionally and frequently, respectively. This suggests nurses working in the current study area were less likely to use EBP than in other countries. It may be due to the nurse's tendency to use more traditional resources such as colleagues and other medical professionals rather than up-to-date resources such as electronic resources and journals to make clinical decisions.

This result was also confirmed by the in-depth interview, *"Since there are no guidelines available for practicing evidence-based nursing in my workplace, I am not doing my job based on the facts that are supported by updated best evidence, and I can only do my job by sharing the problems with my colleague's opinions." (G STA > 10) (F1)*

*"I only work with the knowledge that I have gained from my education, and sometimes I use Google to update my practice records."* (M1) (G STA 5–10)

*"No hospital library, updated guidelines, motivation, enough training, and no computers for nurses to update themselves."* (F2) (G STA 5–10)

The odds of EBP among nurses with MSc and above were 2.148 times higher (AOR = 2.15, 95%CI = (1.15–4.02)) when compared to BSc nurses (see Table 3 above). This is a contrast with the study of Nigeria [9] in which professional qualification has no relationship with the use of good EBP but is analogous with the finding of a study done in Ethiopia [16]. It suggests that good EBP was most often used by nurses who had higher qualifications than by those with lower qualifications. This may be because the educational levels of MSc and above are more technically oriented, improving quest techniques, or being more open to the introduction of EBP into their curricula and teaching programs.

The attitude of nurses towards the integration of evidence-based practice into their clinical practice accounted for 72%. This finding was similar to the study conducted in South Africa (75%) and Singapore (64%). In our finding, the odds of EBP among nurses having a favorable attitude were 1.80 times higher (AOR = 1.80, 95%CI = (1.24–2.62)) when compared to unfavorable attitudes. This finding agrees with the studies conducted in South Africa [22]. This may be due to the difference in nurses' qualification level, work experience, and training, which all have the potential to significantly enhance beliefs and attitudes about EBP. Nurses with a higher level of qualification have better knowledge, which improves their attitude towards EBP.

This result was also confirmed by the in-depth interview, in which most of the respondents suggested that *"it is important for quality care, but the workload, lack of knowledge, and training make us follow previous experience or rely on the opinions of experts."*

**Table 3. Bi-variate and multivariable analysis of evidence-based practice among nurses of the University of Gondar and Tibebe Gion teaching specialized hospitals, Ethiopia 2020 (n = 507).**

| Variables | EBP | | COR (95% CI) | AOR (95%CI) | P-value |
|---|---|---|---|---|---|
| | Poor | Good | | | |
| **Gender** | | | | | |
| Male | 114 | 104 | 1 | 1 | |
| Female | 164 | 125 | 0.695(0.488–0.990) | 1.39(0.95–3.03) | 0.091 |
| **Educational level** | | | | | |
| BSc | 250 | 197 | 1 | 1 | |
| MSc and above | 18 | 42 | 2.961(1.653–5.304) | **2.15(1.15–4.02)**\* | 0.017 |
| **Availability of preferred information sources** | | | | | |
| Available | 240 | 226 | 2.03(1.03–4.01) | 0.86(0.40–1.86) | 0.71 |
| Not available | 28 | 13 | 1 | 1 | |
| **Computer access at the workplace** | | | | | |
| No | 141 | 127 | 1 | 1 | |
| Yes | 94 | 145 | 1.713(1.203–2.438) | 1.032(0.679–1.569) | 0.888 |
| **Internet access at the workplace** | | | | | |
| No | 144 | 93 | 1 | 1 | |
| Yes | 123 | 146 | 1.838(1.29–2.619) | 0.968(0.598–1.566) | 0.894 |
| **Administrative support** | | | | | |
| Not supportive | 218 | 159 | 1 | 1 | |
| supportive | 50 | 80 | 2.194(1.458–3.30) | **1.89(1.22–2.91)**\* | 0.004 |
| **Self-reported availability of enough time** | | | | | |
| No | 158 | 126 | 1 | 1 | |
| Yes | 110 | 113 | 1.288(0.906–1.831) | 0.947(0.630–1.422) | 0.791 |
| **Autonomy-to-change practice by EBP** | | | | | |
| No | 159 | 116 | 1 | 1 | |
| Yes | 109 | 123 | 0.87(0.75–1.01) | 1.19(0.81–1.76) | 0.376 |
| **Knowledge about EBP** | | | | | |
| Poor | 180 | 88 | 1 | 1 | |
| Good | 146 | 93 | 1.303(0.905–1.875) | 0.956(0.636–1.44) | 0.830 |
| **Attitude towards evidence-based practice** | | | | | |
| Unfavorable | 174 | 109 | 1 | 1 | |
| Favorable | 94 | 130 | 2.208(1,544–3.156) | **1.80(1.24–2.62)**\* | 0.002 |
| **Information searching Skills for evidence-based practice** | | | | | |
| Poor | 158 | 94 | 1 | 1 | |
| Good | 110 | 145 | 2.216(1.552–3.162) | 1.385(0.904–2.121 | 0.134 |
| **Preference/use of available Health Information Sources** | | | | | |
| Not preferred | 156 | 92 | 1 | 1 | |
| Preferred | 112 | 147 | 2.226(1.559–3.178) | **2.32(1.58–3.39)**\*\* | 0.000 |
| **Awareness about electronic Health Information Sources** | | | | | |
| Not aware | 148 | 106 | 1 | 1 | |
| Aware | 120 | 133 | 1.547(1.090–2.198) | 1.142(0.76–1.71) | 0.520 |

EBP: Evidence based practice, COR: Crude odds ratio, CI: Confidence interval, AOR: adjusted odds ratio, 1: Reference category

\*: $0.001 < p < 0.05$

\*\*: $p < 0.001$.

The odds of EBP among nurses having administrative support were 1.89 times higher (AOR = 1.89, 95%CI = (1.22–2.91)) when compared to having no support. This is in line with the study conducted in Ethiopia [10]. Other studies also identified managers as key factors, not only for the generation and implementation of EBCP but also for the creation of a good research environment [23].

It was confirmed by the qualitative as:

The majority of participants believed that *"training and budget support for evidence were required."*

*"A training center has to be established to give us training on how to integrate research evidence into our practices and to develop guidelines for practice, thus serving as a source for guiding our health interventions."* (F3) (G STA5)

*"We clinical staff require the assistance of our higher administrators in conducting clinical research because we know our problem better than academic staff, and if we are allowed to research our clinical issue, we can do it and incorporate research evidence into our daily practices."* 5–10 (G STA) (M2)

The odds of EBP among nurses' preferred available information sources were 2.32 times higher (AOR: 2.32, 95%CI: 1.58–3.39) when compared to their counterparts. This study is consistent with prior studies [24, 25], which found that human information sources and sources of printed information were used more frequently by them. This was presumably because of their higher usability and the ease of using them, and because having less Internet access lowers device use and participation in online activities. It was worth noting that the use of electronic information sources by nurses was the least common, given the fact that they have low ability, interpersonal and technical competence in information searching skills and the latest research information.

It was confirmed as:

The majority of nurses said, *"We need training on how to use evidence from journals and databases and how to search for literature to provide effective interventions in health care."*

*"Their desire to use the information is as poor as possible because the profession does not do as much as the profession allows. The nurse professional does not have a culture of communicating and learning from one another. So, we need training on how to use evidence and how to search for literature from databases to provide effective health care interventions."* (G STA greater than 10)(M3)

## Strength

- This study used both quantitative and qualitative research designs to analyze some factors associated with EBP

- A pretest was conducted to test the tools and overall procedures of the study

- Unlike other studies, it has a 95.7% response rate

- This study was conducted in two of the Amhara regional state teaching specialized hospitals (the University of Gondar and Tibebe Gion), which enables the generalizability of the results to other teaching specialized hospitals in Ethiopia.

### Limitations

- There was no more previously studied research in Ethiopia to compare in a local setting.

### Implications of the study

This study's findings have ramifications for practice, education, policy, and research. It will be critical for policymakers and, more specifically, nursing and healthcare professionals to provide high-quality healthcare to patients and their families as a whole.

## Conclusions

According to the findings, only a small number of nurses used EBP to its full potential. The implementation of evidence-based practice was associated with educational level, attitude towards EBP, administrative support, and the availability of information resources, and it was supported by the qualitative investigation.

## Recommendations

There must be a program in place to facilitate the implementation of evidence-based nursing practice by the respective stakeholders (Ethiopian Federal Ministry of Health, Amhara Regional Health Bureau, educators of nursing education, and hospital administration of Amhara Region Teaching Referral public hospitals).

Training should be conducted for nurses on the implementation of evidence-based practice by Amhara Region Teaching Referral public hospitals in collaboration with NGOs, the Ethiopian Federal Ministry of Health, and the Amhara Regional Health Bureau. The resources needed to adopt evidence-based nursing practice should be provided by hospital administration in collaboration with the Ethiopian Federal Ministry of Health, the Amhara Regional Health Bureau, and NGOs. The nursing administration of the research setting should conduct intervention programs involving organizational communication with issues of evidence-based practice implementation in collaboration with nurses to build supportive staff. Given the gap in attitudes toward evidence-based nursing practice, curriculum developers in Ethiopian nursing education programs, particularly in the undergraduate nursing education curriculum, should take evidence-based practice principles into account.

## Supporting information

**S1 Appendix. Research questionnaire.**
(DOCX)

**S2 Appendix.**
(CSV)

**S3 Appendix.**
(CSV)

**S1 File.**
(DOCX)

## Acknowledgments

The authors acknowledge the Institute of Public Health, College of Medicine and Health Science, University of Gondar. Our thanks are also extended to the participating nurses for their cooperation in this study.

## Author Contributions

**Investigation:** Abebe Birhanu Degu.

**Writing – original draft:** Abebe Birhanu Degu.

**Writing – review & editing:** Abebe Birhanu Degu, Tesfahun Melese Yilma, Miftah Abdella Beshir, Anushia Inthiran.

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
