## [Decision Letter · Decision Letter 0]

28 Dec 2021

PONE-D-21-35844Evidence-Based Practice and its Associated Factors among Point-of-care Nurses working at the Teaching and Specialized Hospitals of Northwest Ethiopia: A Concurrent StudyPLOS ONE

Dear Dr. degu,

Thank you for submitting your manuscript to PLOS ONE. After careful consideration, we feel that it has merit but does not fully meet PLOS ONE’s publication criteria as it currently stands. Therefore, we invite you to submit a revised version of the manuscript that addresses the points raised during the review process. Dear Authors, both the reviewers has given positive remark for the manuscript, however, they have suggested minor changes, kindly incorporate the changes and resubmit. Please submit your revised manuscript by Feb 11 2022 11:59PM. If you will need more time than this to complete your revisions, please reply to this message or contact the journal office at plosone@plos.org. Please include the following items when submitting your revised manuscript:A rebuttal letter that responds to each point raised by the academic editor and reviewer(s). You should upload this letter as a separate file labeled 'Response to Reviewers'.A marked-up copy of your manuscript that highlights changes made to the original version. You should upload this as a separate file labeled 'Revised Manuscript with Track Changes'.An unmarked version of your revised paper without tracked changes. You should upload this as a separate file labeled 'Manuscript'.

We look forward to receiving your revised manuscript.

Kind regards,

Rohit Ravi, Ph.D.

Academic Editor

PLOS ONE

Journal Requirements:

2. Thank you for submitting the above manuscript to PLOS ONE. During our internal evaluation of the manuscript, we found significant text overlap between your submission and the following previously published works, some of which you are an author.

- https://www.researchsquare.com/article/rs-477800/v1

- https://www.longdom.org/open-access/evidencebased-practice-utilization-and-associated-factors-among-nurses-working-in-public-hospitals-of-jimma-zone-southwest-ethiopi-25521.html

- https://article.sciencepublishinggroup.com/html/10.11648.j.ajns.20150403.15.html

- https://academicjournals.org/journal/IJNM/article-full-text/11482CC57095

- https://www.researchsquare.com/article/rs-477800/v1

Please revise the manuscript to rephrase the duplicated text, cite your sources, and provide details as to how the current manuscript advances on previous work. Please note that further consideration is dependent on the submission of a manuscript that addresses these concerns about the overlap in text with published work.

3. Please include your tables as part of your main manuscript and remove the individual files. Please note that supplementary tables (should remain/ be uploaded) as separate "supporting information" files.

4. We suggest you thoroughly copyedit your manuscript for language usage, spelling, and grammar. If you do not know anyone who can help you do this, you may wish to consider employing a professional scientific editing service. 

Reviewers' comments:

Reviewer's Responses to Questions

**Comments to the Author**

1. Is the manuscript technically sound, and do the data support the conclusions?

Reviewer #1: Yes

Reviewer #2: Partly

2. Has the statistical analysis been performed appropriately and rigorously? 

Reviewer #1: Yes

Reviewer #2: Yes

3. Have the authors made all data underlying the findings in their manuscript fully available?

Reviewer #1: Yes

Reviewer #2: Yes

4. Is the manuscript presented in an intelligible fashion and written in standard English?

Reviewer #1: Yes

Reviewer #2: Yes

5. Review Comments to the Author

Reviewer #1: The comments have been provided in the attachment. It is a good study to be conducted. In overall, the manuscript is too long. There are parts that can be removed and not too important to explain in manuscript. The manuscript also need some grammar check. There are typos in spelling and certain sentences need to rephrase to make it more clear.

Reviewer #2: Minor revision required with giving space to reference suggested.

More specific conclusion

mention the recommendation separately

Add the suggested reference in your work

Research methodology need to revise in terms of more specific manner

6. PLOS authors have the option to publish the peer review history of their article (what does this mean?). If published, this will include your full peer review and any attached files.

Reviewer #1: No

Reviewer #2: **Yes: **Dr. Rajesh Kumar

---

## [Author Response · Author response to Decision Letter 0]

21 Mar 2022

Thank you all the Academic editor and reviewers for your valuable suggestions and comments.

---

## [Editor Report · Decision Letter 1]

7 Apr 2022

Evidence-Based Practice and its Associated Factors among Point-of-care Nurses working at the Teaching and Specialized Hospitals of Northwest Ethiopia: A Concurrent Study

PONE-D-21-35844R1

Dear Dr. degu,

We’re pleased to inform you that your manuscript has been judged scientifically suitable for publication and will be formally accepted for publication once it meets all outstanding technical requirements.

Kind regards,

Rohit Ravi, Ph.D.

Academic Editor

PLOS ONE

Additional Editor Comments (optional):

Dear authors, the comments by reviewers has been addressed satisfactorily.
---

## [Editor Report · Acceptance letter]

25 Apr 2022

PONE-D-21-35844R1 

Evidence-Based Practice and its Associated Factors among Point-of-care Nurses working at the Teaching and Specialized Hospitals of Northwest Ethiopia: A Concurrent Study 

Dear Dr. Degu:

I'm pleased to inform you that your manuscript has been deemed suitable for publication in PLOS ONE. Congratulations! Your manuscript is now with our production department. 

Kind regards, 

on behalf of

Dr. Rohit Ravi 

Academic Editor

PLOS ONE